# Is the Thickness of the Margin Associated with Local Recurrence and Survival in Patients with Undifferentiated Pleomorphic Sarcoma?

**DOI:** 10.3390/medicina61101881

**Published:** 2025-10-20

**Authors:** Alparslan Yurtbay, Furkan Erdoğan, Bedirhan Albayrak, Ferhat Say, Yakup Sancar Barış, Nevzat Dabak

**Affiliations:** 1Department of Orthopaedics and Traumatology, Samsun University, Samsun 55060, Turkey; 2Department of Orthopaedics and Traumatology, Ondokuz Mayıs University, Samsun 55100, Turkey; furkan.erdogan@omu.edu.tr (F.E.); ferhatsay@gmail.com (F.S.); ndabak@gmail.com (N.D.); 3Department of Orthopaedics and Traumatology, Samsun Research and Training Hospital, Samsun 55090, Turkey; bedirhanalbayrak.5595@hotmail.com; 4Department of Pathology, Ondokuz Mayıs University, Samsun 55100, Turkey; sancarbaris@omu.edu.tr

**Keywords:** orthopedic oncology, soft tissue sarcoma, surgical margin, neoplasm recurrence, disease-free survival

## Abstract

*Background and Objectives*: Undifferentiated pleomorphic sarcoma (UPS) is a rare but aggressive soft tissue tumor. Achieving a tumor-free surgical margin is believed to be crucial for reducing local recurrence; however, the effect of margin thickness on recurrence and survival remains a matter of controversy. This study aimed to assess the association between histopathological margin status and outcomes, including local recurrence, disease-free survival (DFS), and overall survival (OS), in patients with UPS. *Materials and Methods*: We retrospectively analyzed 69 patients with histologically confirmed UPS, identified from a tertiary university hospital tumor database between January 2010 and December 2023. Patients were grouped by histopathological margin status as follows: >1 mm, ≤1 mm, or positive. Recurrence and survival outcomes were analyzed using Kaplan–Meier estimates, Cox regression, and multivariate logistic regression. Patients who underwent reoperation were also evaluated separately. *Results*: Minimum follow-up was 24 months (mean: 52.2 months). Local recurrence occurred in 21 patients, 20 of whom underwent reoperation. Positive margins were significantly associated with higher recurrence risk compared to the >1 mm group (OR: 17.6; 95% CI: 2.88–107.61; *p* = 0.0019). Although recurrence odds were lower in the ≤1 mm group than in the positive group, this was not statistically significant compared to the >1 mm group (OR: 0.52; 95% CI: 0.076–3.50; *p* = 0.498). In reoperated patients, surgical margin status was significantly associated with local recurrence (*p* = 0.0044), and overall survival tended to be longer in those with margins > 1 mm (67.3 ± 47.7 months) or ≤1 mm (50 ± 28.2 months) compared to positive margins (23.3 ± 17.3 months). A moderate negative correlation was observed between age and DFS (*p* < 0.001, r = –0.495). *Conclusions*: This study highlights the prognostic value of surgical margin status in patients undergoing reoperation for local recurrence. In this unique subgroup, margin status was significantly associated with recurrence risk, and patients with negative margins had improved overall survival compared to those with positive margins.

## 1. Introduction

Soft tissue sarcomas (STSs) are a diverse group of cancers originating from mesenchymal cells and exhibiting distinct pathological and clinical characteristics. STSs are rare tumors, accounting for approximately 1% of adult cancers [1]. They represent a diverse group of cancers originating from the embryonic mesoderm and are categorized into more than 50 histologic subtypes and sites of origin. Undifferentiated pleomorphic sarcoma (UPS), one of the common subtypes among soft tissue sarcomas (STSs), is a high-grade, aggressive tumor, previously known as malignant fibrous histiocytoma (MFH). According to the classification made by the World Health Organization in 2002, MFH was included in the UPS category, which provides for pleomorphic tumors outside of histologically identifiable subtypes [2]. UPS, which is usually seen in the adult age group, requires a complex treatment approach due to its heterogeneous clinical behavior, local invasiveness, and high metastatic potential. Local recurrence may develop in approximately 40% of UPS patients, and distant metastasis may develop in up to 30% [3,4].

The surgical management of soft tissue sarcoma benefits significantly from a collaborative, multidisciplinary approach. By fostering partnerships between orthopedic oncologists and plastic surgery teams in specialized referral centers, we can achieve optimal outcomes. This approach enables the execution of wide oncologic resections while considering both effectiveness and minimizing postoperative complications [5]. The treatment protocol for undifferentiated pleomorphic sarcomas is based on extensive oncological resection combined with adjuvant or neoadjuvant radiotherapy, depending on tumor depth, histological grade, and surgical margin characteristics; systemic chemotherapy is mainly targeted to young patient subgroups with tumor diameters of ≥5 cm, advanced disease, or favorable performance score [6]. However, due to the infiltrative nature of this tumor, the adequacy of surgical margins has long been a controversial issue. In the literature, it has been reported that microscopically positive surgical margins significantly increase the risk of local recurrence. In some series, this situation also affects survival [7,8,9]. However, more precise prognostic evaluations have begun to be made by measuring surgical margin width at the millimeter level in different studies. Surgical margin thickness is crucial in achieving local control, especially in subtypes such as UPS and myxofibrosarcoma, which have an infiltrative character [10,11].

This study aimed to investigate the relationship between the thickness of microscopic surgical margins and local recurrence and survival outcomes in patients with UPS. Additionally, we investigated the prognostic significance of histopathological margins in patients who underwent reoperation and examined clinical and parametric factors that may predict local recurrence through multivariate analysis. Our focus on this specific subgroup, which is underrepresented in existing literature, highlights a distinct and valuable aspect of our research.

## 2. Materials and Methods

### 2.1. Study Design and Setting

This study was conducted as a retrospective, comparative analysis, utilizing a longitudinal database that has been consistently maintained at a single tertiary-care university center. The research aims to evaluate specific therapeutic outcomes over time, drawing comparisons among different treatment approaches. It is classified as a Level III study in terms of evidence, indicating that its findings are derived from well-designed cohort studies rather than randomized controlled trials. This classification highlights the significance of the data while also acknowledging the inherent limitations of observational research.

### 2.2. Patients

We conducted a review of 313 patients diagnosed with soft tissue sarcomas who were treated surgically by a multidisciplinary team specializing in bone and soft tissue tumors between January 2010 and December 2023 (Figure 1). Upon analyzing the pathological reports, we identified 85 patients diagnosed with UPS. Five patients died before undergoing surgery, three during adjuvant chemoradiotherapy, and two during the preoperative planning phase due to causes unrelated to chemoradiotherapy or the disease itself, and were therefore excluded from the study. Additionally, we excluded one patient who had undergone unplanned resections at other institutions for UPS, as well as ten patients whose surgical margins were not reported with numerical detail. This led to the exclusion of 18.8% of the initial cohort. Ultimately, our analysis focused on 69 patients, forming the foundation of this study, which aimed to achieve a curative outcome through complete (expanded) resection of localized primary tumors in the extremities. Histological diagnoses and treatment strategies were established through multidisciplinary discussions, following thorough reviews of histopathology and radiology reports. We collected data on tumor grades and surgical margins from the pathological reports, ensuring that all tumors were surgically resected with wide margins. Among the 69 patients who underwent surgical resection, 48 had negative margins, while 21 had positive margins.

### 2.3. Pathology Protocol

Macroscopic assessment and sampling of soft tissue tumors are conducted in accordance with the comprehensive assessment guide established by the Bone and Soft Tissue Pathology Working Group of the Federation of the Turkish Pathology Societies. This guide adheres to international standards and is regularly updated to reflect advancements in pathology practices.

In the initial stages of evaluation, soft tissue tumor resections are meticulously oriented to facilitate the identification of critical surgical margins, previous surgical interventions (including scars or biopsy tracts), and surrounding anatomic structures integral to the tumor’s context. Each surgical margin is colored with distinct inks to enhance visibility and is subsequently sectioned into 1-cm-thick slices for detailed examination.

The minimum sampling requirement mandates that at least one sample must be obtained for every centimeter of the tumor’s largest diameter. In cases where the tumor exceeds 20 cm, a minimum of 20 separate tissue samples should be collected. It is essential to sample all surgical margins, ensuring at least three samples are retrieved from the closest margins to maximize accuracy. Additional samples should be taken from regions bordering identified anatomical structures, such as nerves, blood vessels, or bones, which might be affected by the tumor. Furthermore, any areas of the tumor exhibiting distinct coloration, texture, or consistency should also be sampled to capture potential variations within the tumor.

The results of the microscopic evaluation of the margins are classified according to the Enneking system, which categorizes them as positive (intralesional), marginal, or wide. In instances of marginal resections, a stage millimetric grid is employed to measure the distance to the tumor, which is reported in millimeters. Distances that fall below 1 mm are noted as being closer than 1 mm, providing critical information for evaluating the adequacy of the resection. This thorough approach to macroscopic assessment and sampling ensures that all relevant tumor characteristics are accurately captured for comprehensive pathological evaluation.

All histopathological diagnoses were made by a single experienced sarcoma pathologist. Tumors containing a myxoid component exceeding 50% were classified as myxofibrosarcoma and excluded. Only cases fulfilling the WHO diagnostic criteria for UPS without significant myxoid stroma were included in the analysis.

### 2.4. Treatments

All treatment decisions for patients were meticulously formulated by integrating findings from both radiology and histopathology, while also incorporating the expert recommendations of the multidisciplinary tumor council. This collaborative approach ensured a comprehensive assessment of each case.

Every patient underwent surgical intervention with the primary goal of achieving a negative surgical margin, which is crucial for minimizing the risk of cancer recurrence. In cases where limb-sparing surgery or amputation was necessary, as well as those requiring adjuvant therapies such as radiotherapy or chemotherapy, decisions were made based on thorough discussions within the council. Specifically, attention was given to patients exhibiting histopathological margins of ≤1 mm, where the risk of residual disease is heightened, and the treatment plan was tailored accordingly. If reoperation was performed, the histopathological margins after reoperation were also assessed in the same manner.

Furthermore, patients who had received positive responses to neoadjuvant therapies, either radiotherapy or chemotherapy, were closely evaluated. Their treatment paths were informed by the council’s deliberations, taking into account the individual’s response and the specific nuances of their condition, thereby ensuring the most effective and personalized approach to their care.

### 2.5. Variables

Patients were categorized into two groups based on bone involvement: those with Bone UPS (UPS-B) and those with UPS in soft tissue (UPS-S). Additionally, they were classified by gender into male and female groups. Age served as another criterion, dividing patients into those under 65 years and those aged 65 and above. Tumor size further segmented patients into two categories: those with tumors smaller than 10 cm and those larger than 10 cm. Localization was categorized as upper extremity or lower extremity. Histopathologic margins were classified into three categories: greater than 1 mm, less than or equal to 1 mm, and positive. Patients were also grouped according to the French Federation of Cancer Centers (FNCLCC) grading system, which includes grades 1, 2, and 3. Further divisions were made based on the presence of amputation and metastasis, as well as the receipt of adjuvant or neoadjuvant chemotherapy or radiotherapy, resulting in four additional groups. Prognosis was assessed across three categories: disease-related mortality, non-disease-related mortality, and living patients. Statistical analyses were conducted to examine the relationship between variables and local recurrence and survival status. After reoperation, histopathologic margins were reclassified into three groups based on size: greater than 1 mm, less than or equal to 1 mm, and positive. These analyses aimed to evaluate their effects on local recurrence, disease-free survival, and overall survival.

### 2.6. Primary and Secondary Study Outcomes

The primary objective of this study was to identify the factors that influence the development of local recurrence in patients diagnosed with UPS, as well as to clarify the relationship between histological margin thickness and local recurrence, particularly after surgical resection. Surgical margin status was assessed microscopically in millimeters from histopathological preparations and classified into three categories: positive (0 mm), ≤1 mm negative, and >1 mm negative. Furthermore, the study examined the impact of newly obtained histological margin values from patients who underwent reoperation on local recurrence and survival outcomes. To identify the most significant predictor of local recurrence, multivariate logistic regression analysis was utilized.

The secondary objectives included exploring the relationship between surgical margin status and various clinical and demographic variables related to DFS and OS. Survival times were calculated in months from the date of diagnosis, and survival analyses were performed using the Kaplan–Meier method, with statistical significance assessed accordingly.

### 2.7. Ethical Approval

Ethical approval for this study was obtained from the Ondokuz Mayıs University clinical research ethics committee (approval number: 2025/19, date: 30 January 2025).

### 2.8. Statistical Analysis

The data collected in our study were analyzed utilizing the IBM SPSS Statistics 25.0 software. For continuous variables, we calculated descriptive statistics, including the mean, standard deviation, and minimum and maximum values. Categorical variables are presented as frequencies and percentages (%). The assumption of normal distribution was assessed using the Kolmogorov–Smirnov and Shapiro–Wilk tests. Normality of distribution was assessed using the Shapiro–Wilk test in subgroups comprising fewer than 50 observations. To compare nonparametric variables between two groups, we employed the Mann–Whitney U test. For comparisons among three or more groups, the Kruskal–Wallis test was utilized. The relationship between categorical variables was analyzed using the Pearson Chi-square test and Fisher’s exact test. Survival analyses were conducted using the Kaplan–Meier method, with differences in survival between groups evaluated through the Log-Rank (Mantel–Cox) test. The relationship between DFS and continuous variables was explored via Pearson correlation analysis. Additionally, a multivariate logistic regression analysis was performed to identify independent variables that predict the likelihood of local recurrence. We used a forced entry (enter) method to avoid potential data-driven bias associated with automated stepwise procedures. Variables were selected a priori based on clinical plausibility and their established prognostic significance in sarcoma literature (margin status, tumor size, amputation status, bone involvement, and adjuvant therapy). Although the limited number of recurrence events raised the risk of overfitting, this approach allowed us to test clinically meaningful variables simultaneously in the model. The model demonstrated a high degree of accuracy and explanatory power, with an accuracy rate of 97.1% and an R^2^ value of 0.899. A significance level of *p* < 0.05 was established for all tests.

## 3. Results

Among a cohort of 313 patients diagnosed with soft tissue sarcoma, 85 were identified as having UPS. Following the application of inclusion and exclusion criteria, 69 patients (36 males and 33 females) were enrolled in the study. The mean age of these patients was 59.1 ± 18 years. Of the participants, 42 were younger than 65 years, while 27 were older than 65. Tumor localization revealed that 45 patients had tumors in the lower extremities and 24 in the upper extremities, with all tumors situated beneath the fascia. The mean size of tumors in the lower extremities was 11.1 ± 6.4 cm^2^, compared to 7.7 ± 4.3 cm^2^ for those in the upper extremities. There was one patient with a superficially located tumor; in all remaining cases, the tumors were located deep beneath the fascia. Size metrics remained consistent across both gender and age groups. For patients with tumors measuring less than 10 cm, the average size was 5.6 ± 1.9 cm, whereas for those with tumors larger than 10 cm, the mean size was 15.3 ± 4.8 cm. Among the participants, 18 exhibited bone infiltration, while 51 did not. The patients were categorized according to FNCLCC grading, with 58 patients rated as grade 3, 8 as grade 2, and 3 as grade 1 (Table 1).

A total of 13 patients had histological margins greater than 1 mm, while 35 patients had margins less than or equal to 1 mm, and 21 patients presented with positive margins. The tumor led to local recurrence in 21 patients, and reoperation was performed on 20 of them. Among these, 16 patients had positive histological margins during their first operation, three had margins less than or equal to 1 mm, and one patient had margins greater than 1 mm. During reoperation, 14 patients had margins exceeding 1 mm, three patients had margins of 3 mm or less, and three patients had positive margins. Additionally, 12 patients underwent amputation. The follow-up period for the patients averaged 52.2 months, ranging from 24 to 276 months. There was no metastasis in 50 patients, while 19 patients experienced metastatic disease. Treatment regimens included 12 patients who received adjuvant chemotherapy (CT), 25 patients who underwent neoadjuvant radiotherapy (RT) only, nine patients who received neoadjuvant CT/RT, and 23 patients who did not receive any form of CT/RT. In total, 24 patients died due to disease, 16 patients died from non-disease-related causes, and 29 patients remained alive. Of the 20 patients who underwent reoperation, histological margins were >1 mm in 14 patients, ≤1 mm in 3 patients, and positive in 3 patients (Table 1).

### 3.1. Relationship Between Margin and Local Recurrence

In our study, histopathological margins were categorized into three groups: microscopic positive, ≤1 mm, and >1 mm. We found that local recurrence was not influenced by age, gender, tumor location, FNCLCC Grade, UPS-B/UPS-S status, metastasis, the presence of CT/RT, or prognosis. Additionally, the results indicated that individuals with positive histopathological margins faced a higher risk of local recurrence compared to those with margins >1 mm (OR: 17.6; 95% CI: 2.88–107.61; *p* = 0.0019). Although the odds of recurrence were lower in patients with margins compared to those with margins ≤ 1 mm than those with positive margins, this difference was not statistically significant when compared to the >1 mm group (OR: 0.52;95% CI: 0.076–3.50; *p* = 0.498). When patients were stratified according to resection margin status after reoperation (>1 mm, ≤1 mm, and positive margins), a significant difference was observed in terms of local recurrence rates. Local recurrence occurred in 2 of 14 patients with >1 mm margins, in none of the three patients with ≤1 mm margins, and in all three patients with positive margins. Chi-square analysis demonstrated a statistically significant association between resection margin status and local recurrence (χ^2^ = 10.86, *p* = 0.0044). These findings indicate that positive margins after reoperation are strongly associated with local recurrence, whereas margins of ≤1 mm appeared sufficient to prevent recurrence in this cohort. In particular, Fisher’s exact test comparing the >1 mm and positive surgical margin groups revealed a significantly higher recurrence rate in the positive margin group (*p* = 0.0147). Furthermore, it was observed that the risk of local recurrence was greater in patients with tumors sized less than 10 cm, implying that local recurrence was independent of tumor size (Table 1).

### 3.2. Relationship Between Margin and Overall Survival

The mean overall survival was found to be 52.2 months (95% CI: 22.8 to 39.4). The Kaplan–Meier estimates for overall survival rates at 1, 2, and 5 years were 78.6%, 61.5%, and 38.4%, respectively (Figure 2). The analysis indicated that survival time was statistically independent of factors such as age, gender, tumor size, location, the presence of metastasis, CT/RT status, histopathological margin, and histopathological margin after reoperation (Table 1). However, while not statistically significant, it was noted that overall survival was higher in patients with a histological margin greater than 1 mm compared to those with a margin of 1 mm or less and those with a positive margin (60.2 ± 24.1,) months vs. 52.1 ± 12.4, months vs. 47.4 ± 24.1 months; *p* = 0.418) (Figure 1). Among the 20 patients who underwent reoperation, there was also a trend towards improved overall survival in those with a margin of 1 mm or less compared to those with a margin greater than 1 mm, and both groups compared to those with positive margins (67.3 ± 47.7 months vs. 50 ± 28.2 months vs. 23.3 ± 17.4 months; *p* = 0.841). Among patients who underwent reoperation, the 1st-, 2nd-, and 5th-year survival rates were 93%, 57%, and 14%, respectively, in those with a surgical margin >1 mm; 100%, 67%, and 0% in those with a margin ≤ 1 mm; and 50%, 25%, and 0% in those with a positive margin. These findings indicate that the width of the reoperative histological margin has a direct impact on long-term survival (Figure 3). A statistically significant difference in histological margin post-reoperation was found (*p* = 0.044), although subsequent post hoc analysis did not reveal significant disparities between groups. However, local recurrence was statistically significantly more frequent in tumors smaller than 10 cm in diameter. Overall survival was significantly longer in tumors smaller than 10 cm (Table 1).

### 3.3. Relationship Between Disease-Free Survival and Other Factors

In our study, no significant differences were found regarding disease-free survival (DFS) when considering factors such as histological margin, gender, location, UPS-B/UPS-S, CT/RT, and local recurrence. Although a statistically significant difference in DFS was noted related to the histological margin following reoperation (*p* = 0.049), post hoc analysis revealed no significant differences between the groups. Furthermore, we observed a moderate correlation between age and DFS (*p* < 0.001, r = −0.495).

### 3.4. Logistic Regression Analysis

The factors influencing the occurrence of local recurrence were assessed using multivariate logistic regression analysis. Several covariates were examined, including histological margin status, reoperation histological margin, tumor size, amputation status, and the administration of adjuvant radiotherapy or chemotherapy, as well as the presence of bone involvement. None of these variables achieved statistical significance. However, a reoperation histological margin greater than 1 mm showed a protective trend against local recurrence (OR = 0.17, *p* = 0.109), although this finding did not reach statistical significance.

## 4. Discussion

Numerous studies have been conducted to analyze the various factors influencing local recurrence and prognosis in the treatment of Undifferentiated Pleomorphic Sarcoma (UPS), a condition previously referred to as Malignant Fibrous Histiocytoma (MFH) [4,7,8,9,10,12,13,14,15,16]. These studies often aim to predict patient survival times based on different treatment modalities and clinical characteristics. Our research specifically focuses on evaluating the survival outcomes of patients who experience local recurrence and subsequently undergo reoperation, thereby setting it apart from existing studies in the literature.

Despite the overall low prevalence of UPS, gathering accurate epidemiological data poses significant challenges. This is primarily due to the complex nature of the disease, which leads to variability in its reported incidence. A notable study by Ma et al., published in 2024, highlights that the interpretation of UPS incidence can vary dramatically due to coding discrepancies in its classification within SEER data, further complicating statistical assessments [17]. This variability contributes to the observed heterogeneity in case numbers reported across different research initiatives, thereby complicating direct comparisons of treatment outcomes between institutions (Table 2).

In light of these complexities, our study aims to elucidate the survival outcomes of patients who experience local recurrence, with the goal of providing actionable insights that can inform and enhance management and treatment strategies for this challenging sarcoma.

Research has indicated that the local recurrence rate for soft tissue sarcomas can be alarmingly high, reaching up to 79% in some cases [18,19]. Among the various types of soft tissue sarcomas, myxofibrosarcoma and undifferentiated pleomorphic sarcoma are particularly notorious for their invasive tendencies, as evidenced by their histopathological characteristics. These tumors not only exhibit a significant propensity for local recurrence but also contribute to more complex treatment challenges [8,20]. The implications of local recurrence extend beyond the initial treatment phase; the presence of recurrence is associated with a detrimental impact on patient survival rates. Therefore, it becomes crucial to identify and understand the factors that can effectively mitigate the risk of local recurrence. Furthermore, recognizing the determinants that influence long-term survival serves as a fundamental aspect of developing optimal treatment strategies. These strategies must be individualized, taking into account the specific tumor characteristics and the patient’s overall health, to enhance both outcomes and quality of life.

### 4.1. Relationship Between Margin and Local Recurrence

There are variations among studies regarding the evaluation of surgical margins in UPS, and these differences may account for the discrepancies in local recurrence rates. Some studies focus on the impact of surgical macroscopic margins, while others assess R0, R1, and R2 margins based on measurements of 1–4 mm or 1 cm in diameter. For instance, Fujiwara et al. demonstrated that the risk of local recurrence diminishes in patients with margins greater than 1 cm [8]. Additionally, Junior found that male patients exhibited a higher risk of local recurrence in a series of 42 individuals [12]. Kamat’s research indicated that a positive surgical margin significantly elevates the risk of local recurrence and that radiotherapy is effective in achieving local disease control [7]. In our study, we observed an increased risk of local recurrence in patients with positive histopathological margins. Interestingly, contrary to some findings in the literature, we found that the risk of local recurrence was higher in patients with tumor sizes smaller than 10 cm, suggesting that local recurrence is independent of tumor size. This finding may be attributed to several factors, including the predominance of smaller tumors in anatomically challenging non-thigh locations, a higher—though not statistically significant—rate of positive margins after the initial surgery compared to tumors >10 cm, and the fact that four patients with tumors >10 cm underwent amputation at the initial presentation.

### 4.2. Relationship Between Overall Survival and Affected Factors

Prognostic factors influencing survival in patients with UPS have garnered attention in various studies. In Junior’s research, tumor size exceeding 15 cm, the presence of metastasis, and American Joint Committee on Cancer (AJCC) stage 4 were identified as indicators of poor prognosis [12]. Vodanovich found that larger tumor size, advanced age, metastasis, and positive surgical margins significantly detrimentally impacted survival [9]. Chen’s study highlighted that locally aggressive behaviors, such as recurrence and neural invasion, were determinants of poor prognostic outcomes. Meanwhile, Roland demonstrated that PTEN loss and AXL expression were biologically linked to reduced survival rates [4,15]. According to Makris, factors such as a younger age (<61 years), smaller tumor size (<10 cm), negative surgical margins, and the presence of sporadic tumors are associated with improved survival [14]. Kobayashi et al. demonstrated that adjuvant chemotherapy improved survival, particularly in patients with UPS and tumor size ≥ 10 cm, with the most pronounced effect observed in tumors measuring 10–<15 cm. In contrast, the lack of benefit in patients with tumors <5 cm highlights the critical role of tumor size in determining chemotherapy efficacy [6]. In addition, in our center, the tendency to administer combined chemotherapy/radiotherapy mainly to patients with positive surgical margins may explain the lower overall survival observed in this subgroup. In our study, we observed that survival time was statistically independent of age, gender, tumor size, location, presence of metastasis, KT/RT status, and histopathological margins following reoperation. Additionally, Gusho’s extensive series highlighted that the UPS-b subtype is associated with a more adverse long-term survival trajectory compared to UPS-s [16]. Our research found no significant difference in survival between bone and non-bone UPS groups. Larger cohorts and prospective studies are needed.

### 4.3. Relationship Between Disease-Free Survival and Affected Factors

The key factors influencing DFS in patients with UPS include surgical margin status, microscopic tumor extension, tumor size, grade, and age. Kamat et al. reported a 5-year DFS of 63% for patients with negative surgical margins and only 33% for those with positive margins [7]. Song et al. identified a significant reduction in DFS for patients whose microscopic extensions exceeded 5.5 mm [10]. Furthermore, Fujiwara’s study revealed that the 10-year local recurrence rate decreased to 3% when surgical margins were ≥10 mm, whereas it increased to 25% for zero margins [8].

Chen et al. established a significant relationship between DFS and factors such as age, the quality of surgical margins, and tumor grade [4]. Vodanovich found that the risk of metastasis was 6.8 times higher in tumors larger than 10 cm, and DFS noticeably declined in cases with positive margins [9]. Additionally, it has been noted that adjuvant radiotherapy has a beneficial effect on DFS by reducing metastasis in certain cases. These insights underscore the importance of surgical adequacy and appropriate adjuvant treatment strategies for enhancing DFS in UPS.

In the study by Miwa et al., 10 out of 39 patients (25.6%) who received neoadjuvant chemotherapy showed a good response (≥90% tumor necrosis), and these responders had a significantly higher metastasis-free survival rate compared to non-responders (*p* = 0.034) [21]. In our study, although not statistically significant, patients who received adjuvant radiotherapy had a higher mean DFS. However, this may be attributed to the higher frequency of radiotherapy administration in patients with surgical margins ≤ 1 mm, specifically in those with RT (63.1%), KT (41.5%), KT/RT (37.5%), and those with no treatment (51.7%).

In our study, we did not observe any significant differences in DFS concerning histological margins, gender, tumor location, bone versus non-bone UPS, KT/RT methods, or the presence of local recurrence. However, a statistically significant difference in DFS was noted with histological margins following reoperation (*p* = 0.049), although post hoc analyses did not reveal significant differences between groups. Notably, a statistically moderate correlation was found between age and DFS (*p* < 0.001, r = −0.495).

Although none of the variables retained statistical significance in our logistic regression model, this limitation is most likely attributable to the restricted number of recurrence events in our cohort. Nevertheless, the observed protective trend of a reoperation margin > 1 mm (OR = 0.17, *p* = 0.109) suggests a potential clinical benefit that warrants further validation in larger patient populations. Similarly, the absence of significant associations for tumor size, amputation status, adjuvant therapy, and bone involvement should be interpreted with caution, as the study may have been underpowered to detect smaller effect sizes. These findings highlight the need for larger multicenter studies with sufficient statistical power to more accurately identify independent predictors of local recurrence in UPS.

### 4.4. Limitations and Strengths

This study presents several limitations that deserve consideration. Firstly, its retrospective design and single-center approach restrict the generalizability of the findings, while the relatively small cohort size may hinder statistical power, especially in subgroup analyses. Secondly, molecular prognostic markers such as PTEN or AXL, which are emphasized in recent literature, were unavailable for our patients and, therefore, could not be incorporated into the study.

Despite these limitations, our study possesses significant strengths. To our knowledge, it is among the few reports specifically examining reoperation cases of high-grade undifferentiated pleomorphic sarcoma (UPS) with detailed millimeter-level margin measurements. All pathological evaluations followed standardized national guidelines, ensuring consistency and reliability in margin assessment. Furthermore, we conducted comprehensive multivariate analyses and provided long-term follow-up data, which offer clinically relevant insights regarding the prognostic value of surgical margin thickness in this rare and aggressive tumor.

## 5. Conclusions

This study illustrates that histopathological assessment of surgical margin status is significantly associated with local recurrence, particularly following reoperation, in patients diagnosed with UPS. While positive surgical margins do raise the risk of local recurrence, surgical margin status alone does not independently impact OS or DFS. Among the various factors assessed, only age demonstrated a statistically significant and moderately negative correlation with DFS. These findings indicate that achieving wider negative surgical margins, especially for patients undergoing reoperation, may help reduce local recurrence. However, it is essential to acknowledge that survival outcomes depend not only on surgical margins but also on a multitude of clinical and biological factors. Larger, multicenter, and prospective studies are necessary to more precisely define optimal surgical strategies and prognostic markers for patients with UPS.

## Figures and Tables

**Figure 1 medicina-61-01881-f001:**
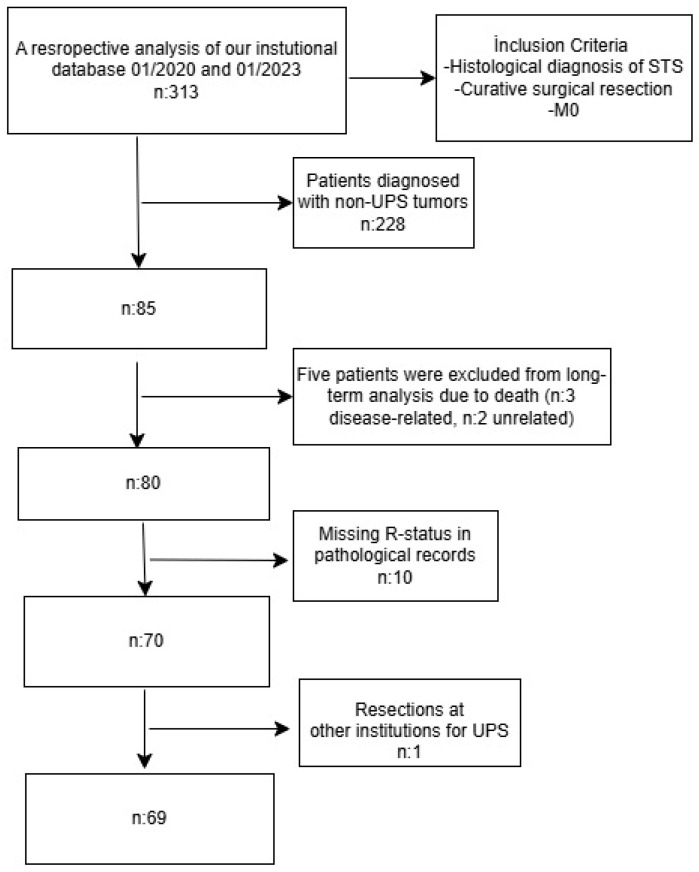
Flow diagram of inclusion and exclusion criteria.

**Figure 2 medicina-61-01881-f002:**
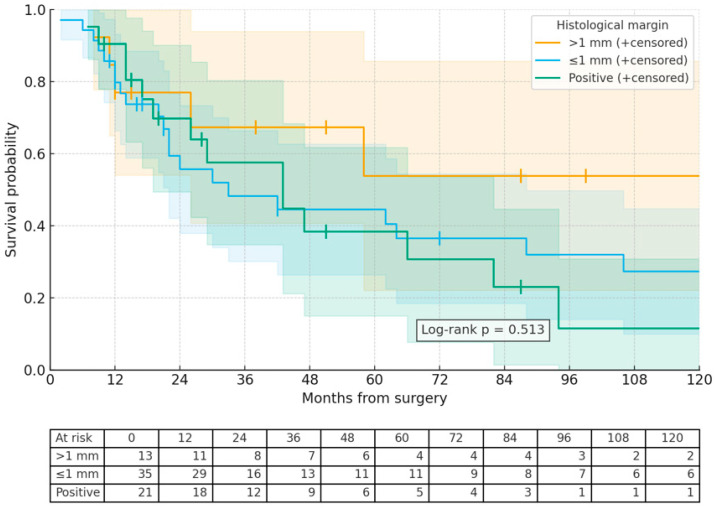
Cumulative survival rates were compared according to histopathological margin groups.

**Figure 3 medicina-61-01881-f003:**
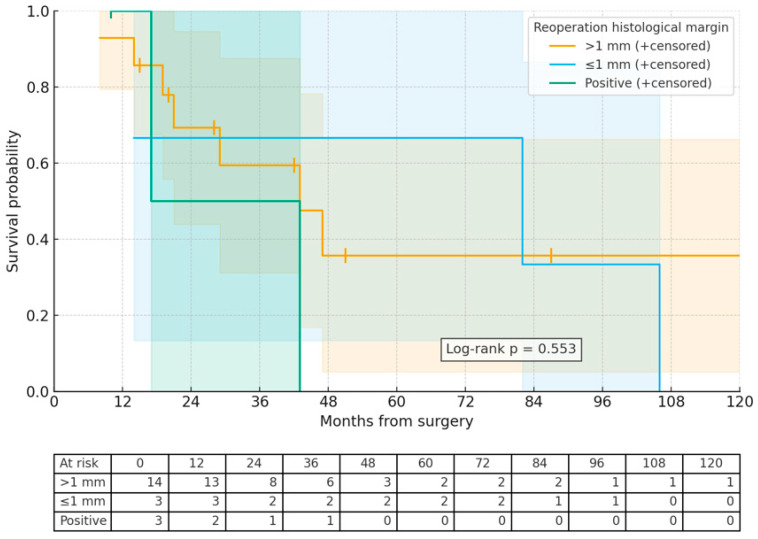
Cumulative survival according to histopathological margin after reoperation.

**Table 1 medicina-61-01881-t001:** Relationship Between Local Recurrence, Overall Survival Time, and Parameters.

	Local Recurrence (*n*, %)		*p* Value	Overall Survival Time (Mean ± SD, Months)	*p* Value
	No	Yes			
**Age**					
<65 years	30 (71.4)	12 (28.2)	0.783	67.2 ± 71.7	0.413
>65 years	18 (66.7)	9 (33.3)		28.9 ± 24.8	
**Gender**					
Women	21 (63.6)	12 (36.4)	0.305	61.9 ± 66.9	0.150
Men	27 (75)	9 (25)		43.3 ± 53.9	
**Tumor Location**					
Upper Extremity	17 (70.8)	7 (29.2)	0.867	29.5 ± 5.2	0.594
Lower Extremity	31 (68.9)	14 (31.1)		34.4 ± 7.3	
**Histopathological Margin**					
>1 mm	11 (84.6)	2 (15.4)	0.0019 *	60.2 ± 24.1	0.841
≤1 mm	32 (91.4)	3 (8.6)		52.1 ± 12.4	
Positive	5 (23.8)	16 (76.2)		47.4 ± 24.1	
**FNCLCC Grade**					
1	2 (66.7)	1 (33.3)	0.545	106.3 ± 142.9	0.255
2	7 (87.5)	1 (12.5)		39.3 ± 38.2	
3	36 (67.2)	19 (32.8)		51.2 ± 57.7	
**Bone involvement**					
UPS-B	11 (61.1)	7 (38.9)	0.365	65.7 ± 52.8	0.238
UPS-S	37 (72.5)	14 (27.5)		47.4 ± 63	
**Tumor Size**					
>10 cm	26 (83.9)	5 (16.1)	0.02 *	35.9 ± 49	0.03 *
≤10 cm	22 (57.9)	16 (42.1)		65.5 ± 66.5	
**Amputation**					
Yes	5 (41.7)	7 (58.3)	0.021 *	71.2 ± 90.3	0.410
No	43 (75.4)	14 (24.6)		48.2 ± 52.7	
**Metastasis**					
Yes	35 (70)	15 (30)	0.899	54.3 ± 60.5	0.659
No	13 (68.4)	6 (31.6)		46.8 ± 62.7	
**Adjuvant/Neoadjuvant Treatment**					
Adj. CT	6 (50)	6 (50)	0.227	41.5 ± 26.8	0.885
Adj. RT	18 (78.3)	5 (21.7)		51.7 ± 56.9	
Neoadj. RT	19 (76)	6 (24)		63.16 ± 82.06	
Neoadj. CT/RT	5 (55.6)	4 (44.4)		37.5 ± 21.9	
**Prognosis**					
D	15 (62.5)	9 (37.5)	0.502	29 ± 21.2	
ND	13 (81.2)	3 (18.8)		34.3 ± 34.2	0.018 *
Alive	20 (69)	9 (31)		81.4 ± 80.3	
**Histopathological Margin** **at Reoperation**					
>1 mm	12 (85.7)	2 (14.3)	0.0044 *	50 ± 28.2	0.684
≤1 mm	3 (100)	0 (0)		67.3 ± 47.7	
Positive	0 (0)	3 (100)		23.3 ± 17.3	

FNCLCC: French Federation of Cancer Centers Sarcoma Group; UPS: Undifferentiated Pleomorphic Sarcoma; UPS-B: Bone Undifferentiated Pleomorphic Sarcoma; UPS-S: Soft Tissue Undifferentiated Pleomorphic Sarcoma; D: Disease-related death; ND: Non-disease-related death; CT: Chemotherapy; RT: Radiotherapy; mm: Millimeter; SD: Standard Deviation; *p*: Probability value. * Statistically significant (*p* < 0.05).

**Table 2 medicina-61-01881-t002:** Results of surgical margin on local recurrence and survival for patients with UPS.

Author	Number of Patients	Year	Tumor Type	Margin Category	Local Recurrence	Survival (5 Years)	Comment About Local Recurrence	Comment About Poor Prognosis
Fujiwara [8]	305	2020	MFS-UPS	R0, R1, R2	19%	81%	>1 cm surgical margin reduces the risk of local recurrence	---
Maçaneiro Junior [12]	42	2023	UPS	neg.-pos.	14%	25.9 months	The male gender may increase the likelihood of local recurrence	>15 cm size, metastasis, AJCC grade 4 is associated with a poor prognosis
Winchester [13]	319	2018	UPS	dermal, subfascial	Unspecified	Unspecified	---	A 2 cm size, subfascial location is associated with poor prognosis.
Vodanovich [9]	266	2019	UPS	neg.-pos.	---	60%	---	Big tumor size, age, metastasis, and positive surgical margin adversely affect survival.
Chen [4]	100	2019	UPS	R0, R1/R2	40%	53%	---	Tumor size, recurrence, and neurovascular invasion are associated with poor prognosis.
Makris [14]	208	2020	UPS	neg.-pos.	27%	63%	---	age < 61, size < 10 cm, neg. surgical margin, sporadic TM. Associated with improved survival
Kamat [7]	55	2019	UPS (lower extr.)	r0: >4 mm, r1: 1–4 mm., pos.: <1 mm	7%	68%	A positive surgical margin is associated with an increased risk of recurrence. RT improves local control, but doesn’t affect DFS.	---
Song [10]	20	2018	UPS (extr.-trunk)	neg.-pos.	---	A microscopic spread greater than 5.5 mm is a predictor of local recurrence.	---	---
Roland [15]	208	2016	UPS (sporadic/RA)	neg.-pos.	27%	63%	---	Loss of PTEN and AXL expression is associated with poor survival.
Gusho [16]	4729	2022	UPS (Bone and Soft Tissue)	Unspecified	Unspecified	s: %60, b: 51.2	---	The long-term survival rate of UPS-B is worse than that of UPS-S.

AJCC: American Joint Committee on Cancer, DFS: Disease-Free Survival, UPS: Undifferentiated Pleomorphic Sarcoma, UPS-B: Bone Undifferentiated Pleomorphic Sarcoma, UPS-S: Soft Tissue Undifferentiated Pleomorphic Sarcoma, RT: Radiotherapy, Pos.: Positive, Neg.: Negative, Extr.: Extremity, MFS: Myxofibrosarcoma, RA: Radiation-Associated Undifferentiated Pleomorphic Sarcoma, mm: Millimeter.

## Data Availability

The datasets used and analyzed during the current study are available from the corresponding author upon reasonable request.

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
