# Peer review of "Is the Thickness of the Margin Associated with Local Recurrence and Survival in Patients with Undifferentiated Pleomorphic Sarcoma?"

_medicina, 2025, doi:10.3390/medicina61101881_

Round 1

Reviewer 1 Report (New Reviewer)

Comments and Suggestions for Authors

Thank you for submitting your manuscript to the Medicina. This study examines the relationship between surgical margin thickness and clinical outcomes in patients with undifferentiated pleomorphic sarcoma. It addresses a clinically important issue and provides valuable data on reoperation. However, this study has several methodological concerns that limit the interpretability of the findings. So, I suggest revising the manuscript, and here are my suggestions. With appropriate revisions addressing these issues, this manuscript could make a valuable contribution to the sarcoma treatment.

  1. In lines 312-316, the authors provide no justification for variable selection in their multivariate logistic regression model. With only 21 recurrence cases and 6-7 covariates, there is a risk of overfitting. The authors must specify the rationale for including specific variables and statistical approach used (stepwise, forward, backward selection)

  1. Five patients who died within 24 months were excluded, yet this received no discussion in the manuscript. This exclusion likely introduced significant survivor bias, artificially improving survival estimates. I suggest the authors consider acknowledging this limitation in the discussion and explain how this affects the generalizability to the overall UPS population.

  1. There is an ambiguity of histological classification: distinguishing UPS from myxofibrosarcoma (MFH with myxoid stroma portion), despite their distinct clinical behaviors and prognosis. Given myxofibrosarcoma’s myxoid stromal components and higher recurrence rates compared to UPS, the author should clarify the histopathological criteria used for differentiation and inclusion criteria (like, whether myxoid components were evaluated) or, if both subtypes were included, provide separate analysis.

  1. Table 1 formatting inconsistencies (missing units, incomplete values)
  2. Kaplan-Meier curve lack confidence interval and risk tables
  3. In the discussion, the authors should expand their explanation about the counterintuitive finding that smaller tumors (<10 cm) had higher local recurrence, which contradicts established literature. Are there any ideas for potential confounding factors?

Author Response

Reviewer 2 Report (Previous Reviewer 1)

Comments and Suggestions for Authors

Dear Editor, dear Authors,

I appreciate the efforts made by the authors to improve the current version of the manuscript. Several important improvements have been implemented, although some of the previous observations were not addressed. I would like to congratulate the authors for the work they have invested, and I trust that the editor will ultimately decide whether the manuscript is now in a form suitable for publication in such a reputable journal.

As a final observation, I would kindly suggest that the authors avoid beginning line 317 with the word “interestingly.” In addition, part of the text in section "3.4 Logistic Regression Analysis" is overly descriptive of the results and would contribute more meaningfully if incorporated into the Discussion section.

With respect and appreciation,
Reviewer

Author Response

This manuscript is a resubmission of an earlier submission. The following is a list of the peer review reports and author responses from that submission.

Round 1

Reviewer 1 Report

Comments and Suggestions for Authors

Dear Editor, dear Authors,

Thank you for the opportunity to review this manuscript.

I appreciate the authors' efforts in addressing the important topic of the impact of resection margin thickness on the prognosis of patients with undifferentiated pleomorphic sarcoma, particularly in relation to recurrence and survival. The study was approved by the institutional ethics committee of the university.

To develop this study, the authors included 69 patients (out of 313 evaluated) who were retrospectively selected from the database of a tertiary university center between January 2010 and December 2023. The exclusion of patients who did not meet the inclusion criteria is illustrated in the flowchart presented in Figure 1. The 69 patients included in the study underwent complete tumor resections. Among these patients, negative resection margins were identified in 48 cases, while positive margins were observed in 21 cases.

I appreciate the detailed presentation of the statistical methods described in subsection 2.8 – Statistical Analysis. However, I would like to mention that for small sample sizes, typically under 50 subjects, the Shapiro-Wilk test is generally recommended for assessing normality. It is likely the most appropriate option for evaluating the distribution within your subgroups as well [Razali, N.M.; Wah, Y.B. Power comparisons of Shapiro–Wilk, Kolmogorov–Smirnov, Lilliefors and Anderson–Darling tests. J. Stat. Model. Anal. 2011, 2, 21–33.].

A sensitive aspect of this study is the heterogeneity of patients regarding the resection margin distance, which is a key prognostic factor. However, in the context of the pathology under investigation, it is understandable that assembling a large and homogeneous cohort is challenging, even over an extended period of more than 10 years.

For the Kaplan-Meier curves, it would be useful to better emphasize the initial part of the survival timeline, where most of the events occurred. I would suggest adjusting the time axis to display intervals of 12 months, which may improve interpretability. I also noticed a substantial number of censored cases, without predefined events, beyond 100 months. Therefore, you might consider limiting the time axis to 108 months (approximately 9 years), if your statistical software allows this adjustment. If the resulting graph offers a clearer visualization, you may use it in the final version of the manuscript. This suggestion applies to both Figures 2 and 3. Additionally, I would recommend displaying the p value “p = ... – Log-Rank test” directly on the survival plots, ideally in a small text box, to make the figures more complete and facilitate readers' understanding of the differences between groups. Please also verify whether the last item in the legend refers to censored cases among the positive group, as is mentioned only “Positive”.

It would be interesting to include a table presenting both the univariate and multivariate Cox regression models for the variables considered in the analysis (for section 3.4). The covariates included in the multivariate model could be specified in a note below the table. However, it should be noted that conducting such an analysis in small sample groups may be challenging, as the literature recommends at least 10 events per covariate to ensure the validity of the model  [Domburg R, Hoeks S, Kardys I, Lenzen M, Boersma E. Tools and Techniques - Statistics: How many variables are allowed in the logistic and Cox regression models? EuroIntervention : journal of EuroPCR in collaboration with the Working Group on Interventional Cardiology of the European Society of Cardiology. 2014 Apr 23;9:1472–3].

It is recommended that all abbreviations used in figures and tables be clearly explained in the caption or directly below the respective representation. This practice enhances clarity and ensures the content is accessible to a broader audience [Tuncel A, Atan A. How to clearly articulate results and construct tables and figures in a scientific paper? Turk J Urol. 2013 Sep;39(Suppl 1):16–9. doi:10.5152/tud.2013.048]

The citations mentioned in this review are not part of my own work, nor of any collaborative or institutional research. They represent a selection of articles that helped me better understand the field and support my perspective. These references are included solely to substantiate the reviewer’s comments and are not intended to be cited by the authors in their manuscript. The reviewer considers such background documentation an important aspect of producing a high-quality and well-informed review.

I hope this review proves constructive and contributes to the improvement of the manuscript. The suggestions provided are not mandatory; the authors are encouraged to implement those they find reasonable and valuable.

I appreciate the effort the authors have invested in the research and preparation of this work.

With respect and consideration,

Reviewer

Reviewer 2 Report

Comments and Suggestions for Authors

Journal: Medicina

Manuscript Title: Is the Thickness of the Margin Associated with Local Recurrence and Survival in Patients with Undifferentiated Pleomorphic Sarcoma?

Authors: A Yurtbay, F Erdoğan , B Albayrak , F Say , Y Sancar Barış, N Dabak

Manuscript ID: medicina-3756144

This manuscript examines whether the thickness of microscopic surgical margins affects local recurrence and survival in patients with undifferentiated pleomorphic sarcoma (UPS). While the study may be of interest regarding the prognostic value of margin status in UPS patients undergoing reoperation, several methodological and translational limitations should be addressed:

  1. The study’s retrospective nature and reliance on data from a single tertiary center may introduce selection bias and unmeasured confounding factors. Notably, 84% of tumors were high-grade, which may not represent the broader UPS patient population, and the cohort lacks racial and ethnic diversity.
  2. A significant proportion (18.8%) of the initial cohort was excluded due to insufficient follow-up, missing margin data, or prior unplanned resections. The absence of analysis or discussion regarding the characteristics of these excluded patients raises concerns about potential selection bias and the representativeness of the final cohort.
  3. The final sample size of 69 patients restricts the statistical power, especially for subgroup analyses. For example, only three patients had ≤1 mm margins following reoperation.
  4. The study does not adequately adjust for the impact of adjuvant therapies such as radiotherapy and chemotherapy, which varied among patients. Additionally, while tumor size influenced outcomes, it was not prioritized as a covariate in the multivariate analyses, potentially confounding results.
  5. Important molecular markers known to influence UPS prognosis, such as PTEN loss and AXL expression, were not evaluated. Omitting these factors limits the biological depth of the study and its comparability with contemporary research.
  6. The finding that local recurrence was higher in tumors smaller than 10 cm is unexpected and may reflect unaddressed confounding or bias in the analysis.
  7. The results may not be generalizable to broader or more diverse patient populations, as treatment protocols and demographics may differ at other institutions. The lack of external validation further limits the reproducibility and applicability of the findings.
  8. The methods for measuring and classifying surgical margins differ from those commonly used in literature, which may complicate comparisons with other studies and cross-study validation.
  9. The manuscript would benefit from comprehensive editing to improve clarity and readability.
